



# The sensitivity of EC-Earth3 decadal predictions to the choice of volcanic forcing dataset: Insights for the next major eruption.

Roberto Bilbao[1], Thomas J. Aubry[2], Matthew Toohey[3], and Pablo Ortega[1]

[1]Earth Science Department, Barcelona Supercomputing Center (BSC), Barcelona, Spain
[2]Department of Earth and Environmental Sciences, University of Exeter, Penryn, UK
[3]Institute of Space and Atmospheric Studies, University of Saskatchewan, Saskatoon, Canada

**Correspondence:** Roberto Bilbao (roberto.bilbao@bsc.es)

**Abstract.** Large volcanic eruptions can have significant climatic impacts. Due to their unpredictable nature, such eruptions can render operational decadal forecasts inaccurate. To benefit from the strong climate signals they exert, which enhance climate predictability, decadal forecasts must be rerun with updated estimates of the stratospheric sulfate aerosol evolution. Two tools to rapidly generate the volcanic forcings are the Easy Volcanic Aerosol (EVA, Toohey et al., 2016) and its updated version,
EVA_H (Aubry et al., 2020). In order to validate the use of the volcanic forcings generated with these simple models in decadal forecasts, we compare the volcanic forcings generated with EVA and EVA_H with CMIP6 for the recent eruptions of Mount Agung (1963), El Chichón (1982) and Mount Pinatubo (1991) and investigate the consistency in their associated climate responses in decadal predictions produced with the BSC decadal forecast system. Our findings reveal differences in the magnitude and latitudinal structure of the forcings generated by EVA and EVA_H compared to the official CMIP6 forcings,
particularly for the eruptions of Mount Agung and El Chichón. These differences in the volcanic forcings lead to some global and regional quantitative differences in the predicted radiative responses, as evidenced in variables like the top-of-atmosphere (TOA) net radiative fluxes, surface temperature, and lower stratospheric temperature. Despite these differences, comparing the predicted anomalies in those variables with observations, we show that either of the forcings considered allows to make skillful predictions after the major volcanic eruptions. Our study thus supports both EVA and EVA_H generated forcings as reasonable
choices for predicting the post-volcanic radiative responses.

## 1 Introduction

Decadal climate prediction systems have become a useful tool for forecasting the climate of the next few years out to several decades (e.g. Hermanson et al., 2022). Part of the predictability in decadal forecasts arises from initialising the model from the observed state. Initialization is key to take advantage of the slowly evolving components of the climate system (i.e., the ocean),
which can be predictable, and phases the model's internal climate variability with the observed one (e.g. Doblas-Reyes et al., 2013). The other main source of predictability is the use of accurate estimates of external radiative forcings, which can be of natural (e.g., solar irradiance and volcanic aerosols) or anthropogenic (e.g., greenhouse gas concentrations, land use changes and anthropogenic aerosols) origin. Among the natural forcings, volcanic stratospheric sulphate aerosols are one of the main drivers of seasonal to centennial climate variability (e.g. Sigl et al., 2015) and are thus critical for predictability.



Explosive volcanic eruptions inject large quantities of sulphur dioxide into the stratosphere, where sulphate aerosols are formed and transported globally by the atmospheric circulation. The primary effect of the aerosols is to scatter part of the incoming solar radiation back into space, causing a negative radiative forcing that cools the Earth's surface, an effect that may last for several years until the aerosols are transported sediment to the troposphere where they are washed out within weeks (e.g. Robock, 2000). These aerosols also absorb infrared radiation which leads to a local warming of the stratosphere. These temperature adjustments may lead to other climate impacts such as changes in the atmospheric and oceanic dynamics, which modulate climate variability and the global cooling response to volcanic eruptions (see Marshall et al. (2022) and references therein). Studies have shown that volcanic impacts on climate have a high predictive potential on seasonal-to-decadal timescales (e.g. Timmreck et al., 2016; Ménégoz et al., 2018; Hermanson et al., 2020; Bilbao et al., 2024). Therefore, including the volcanic forcing in operational climate forecasts is necessary to produce accurate predictions whenever an explosive volcanic eruption occurs.

In the current operational decadal prediction systems, stratospheric sulphate aerosols are typically prescribed as boundary conditions. Therefore, in a real-time prediction context, should a new major volcanic eruption occur, the estimates of the evolution of the corresponding volcanic aerosols would be needed. The method recommended by the SPARC/DCPP Volcanic Response initiative (VolRes Sospedra-Alfonso et al., 2024) to produce the forcing is to use the Easy Volcanic Aerosol model (EVA Toohey et al., 2016) or its more recent version EVA_H (EVA_H Aubry et al., 2020). These are idealised models developed to reconstruct the spatiotemporal evolution of the stratospheric aerosol optical properties for a given input list of volcanic eruption characteristic attributes. The output can be then formatted accordingly and implemented in climate models.

While these tools, particularly EVA, have been used widely for a variety of studies to investigate the impact of volcanic eruptions on climate, the realism and the underlying uncertainty in the climate predictions associated with the volcanic forcings remains unexplored. The objective of this paper is to evaluate the radiative response to the volcanic forcings produced with EVA and EVA_H using decadal hindcasts for the recent eruptions of Mount Agung (1963), El Chichón (1982) and Mount Pinatubo (1991), which may inform the expected uncertainty if used in real-time forecasts. The paper is structured as follows. Section 2 describes how the EVA and EVA_H forcings are produced and the decadal hindcast run with the BSC decadal forecast system (Bilbao et al., 2021). Section 3 compares the EVA and EVA_H forcings with the CMIP6 historical volcanic forcing and evaluates the simulated climate response to these forcings. Section 4 summarises and discusses the results in the context of operational forecasts.

## 2 Methods

### 2.1 EVA and EVA_H stratospheric aerosol forcings.

Toohey et al. (2016) developed EVA based on a parameterized three-box model of stratospheric transport and simple scaling relationships used to derive mid-visible (550 nm) aerosol optical depth and aerosol effective radius from stratospheric sulphate mass. EVA was calibrated against the Chemistry-Climate Model Initiative (CCMI Eyring and Lamarque, 2012) satellite dataset for the 1991 Pinatubo eruption. To address some of the limitations in EVA, Aubry et al. (2020) developed



an extension named EVA_H (the H stands for height) with the following improvements: i) making the forcing magnitude and its time evolution dependent on the injection altitude as well as latitude; ii) making the vertical structure of the aerosol optical properties and its time evolution dependent on the injection altitude; iii) calibrating the model against the more recent Global Space-based Stratospheric Aerosol Climatology (GloSSAC) observation-based dataset (Thomason et al., 2018) for the full 1979-2015 period. Despite the improvements implemented in EVA_H, one lingering limitation affecting both models is a too slow decay of the aerosol forcing for eruptions that inject 1 Tg SO2 or less into the stratosphere (Vernier et al., 2024). EVA and EVA_H are both publicly available on github (https://github.com/matthew2e/easy-volcanic-aerosol and https://github.com/thomasaubry/EVA_H).

Both EVA and EVA_H, output stratospheric aerosol optical properties as a function of time, latitude, height, and wavelength based on several parameters which describe the eruption characteristics, termed eruption source parameters. EVA uses the mass of sulphur injected, timing of the eruption and an optional hemispheric asymmetry parameter (indicating the degree of asymmetry in the transport of aerosol to the Northern Hemisphere and Southern Hemisphere). EVA_H additionally requires the tropopause altitude at the volcano location and the altitude of the SO2 injection, and does not include an asymmetry parameter. Further parameters to generate the forcings are required, such as the time span of forcing files and the model wavelengths. In the case of EC-Earth3, we calculate 14 shortwave and 16 long wave bands, as used in the IFS radiation scheme. Note that although model parameters such as aerosol loss timescales can be adjusted for both EVA and EVA_H, we use the default recommended parameter values.

EVA and EVA_H produce monthly values of the aerosol extinction (EXT), single scattering albedo (SSA) and scattering asymmetry factor (AF) as a function of time, altitude (up to 40 km), latitude and wavelength, as well as providing the altitude and latitude values. These variables are produced on a standard set of levels and therefore were interpolated to the Integrated Forecast System (IFS; atmospheric model of EC-Earth3) vertical levels (91 vertical levels). The monthly variation of the altitude values is taken into account considering a climatology of the IFS layers height. Values above 40 km height were masked as they can be unrealistically distorted due to the vertical extrapolation. The latitudinal interpolation is directly performed online by EC-Earth3. With these variables, the aerosol optical depth (AOD), aerosol absorption optical depth (AAOD) and the scattering part of the aerosol optical depth (REFAOD) as a function of the waves bands, time, latitude and altitude, are derived and the EC-Earth3 input forcing files generated.

In this work we produce historical volcanic forcings with both EVA and EVA_H for the period 1962-2005 including only the 1963 eruption of Mount Agung, the 1982 eruption of El Chichón and the 1991 eruption of Mount Pinatubo. The eruption source parameters used for EVA are those in Toohey et al. (2016) (Table S1). For EVA_H, the eruption source parameters eruption for Mount Agung were taken from Niemeier et al. (2019), splitting the eruption into two phases as recommended, and for the eruptions of El Chichón and Mount Pinatubo they were taken from Carn et al. (2016) (Table S2). Additionally, we produce another volcanic forcing with EVA_H for the period 1979-2005 including all the eruptions in Carn et al. (2016), we refer to this forcing as EVA_H_Carn16.



## 2.2 Decadal Prediction Experiments, Observations and Evaluation Methods

The experiments analysed in this study were performed with the BSC Decadal Forecast System (Bilbao et al., 2021), which is based on the Coupled Model Intercomparison Project Phase 6 (CMIP6) version of the EC-Earth3 atmosphere-ocean general circulation model (AOGCM) in its standard resolution (∼1° both in the ocean and atmosphere; Döscher et al. (2022)). Our decadal prediction system follows the CMIP6 Decadal Climate Prediction Project (DCPP) component-A protocol (DCPP Boer et al., 2016) and, therefore, consists of 10-member ensembles of 10-year-long predictions initialised every year in November from 1960 to present, with prescribed CMIP6 historical forcings, including the stratospheric volcanic aerosols (Luo, 2018).

To evaluate the climate response to volcanic forcings we follow a protocol similar to the Decadal Climate Prediction Project (DCPP Boer et al., 2016), which was designed jointly with VolMIP (Zanchettin et al., 2016), to investigate the impact of volcanic eruptions on decadal predictions. DCPP component-C repeats the 10-member hindcasts initialised in 1962, 1981 and 1990 (corresponding with the start-dates right before the onset of the three major volcanic eruptions), but without the volcanic forcings, which are then compared with the baseline hindcast (DCPP-A) to diagnose the effects of the forcings. To evaluate the realism of the EVA and EVA_H estimates, we repeat the forecasts for the same three start dates but using instead with their corresponding volcanic forcings. The experiments are summarised in table 1. The impact of the volcanic forcing is computed by subtracting the DCPP-C hindcasts from those with volcanic forcing, i.e., DCPP-A minus DCPP-C. Significance on the differences for both time series and fields is tested by creating a distribution of 10-member mean differences by bootstrap with replacement of ensemble members from 1000 repetitions. If the 2.5%-97.5% range of the distribution of differences (equivalent to p ≤ 0.05) does not include zero we reject the null hypothesis (no difference between the hindcast sets with and without volcanic forcing) and the differences are considered statistically significant. For the time series plots, the uncertainty is shown by the 10th and the 90th percentiles of the hindcast experiments ensemble.

The prediction anomalies were computed using the lead-time-dependent climatology from the DCPP-A decadal hindcasts and we assume that the forecast drift is equal in all the hindcast experiments, which is a reasonable assumption given that large volcanic eruptions are uncommon. To evaluate the hindcasts we use the ERA5 reanalysis (Hersbach et al., 2020) for TOA fluxes, near-surface temperature and lower stratospheric temperature (at 100 hPa). Additionally we use the satellite derived datasets: Deep-C version 5 reconstruction (Liu et al., 2020) for TOA fluxes and the TLS (Temperature Lower Stratosphere) single-channel data set of zonally averaged temperature anomalies, produced by Remote Sensing Systems (RSS Mears and Wentz, 2009). This last dataset is sensitive to temperatures between altitudes of 12 km to 27 km and is derived using vertical weighting functions (see Mears and Wentz, 2009, for further details). For the model experiments not enough vertical model levels were saved for atmospheric temperature to apply a vertical weighting function as in TSL, so we take the temperature at 100 hPa which approximately corresponds with the maxima.

To quantify the errors in the hindcasts for the ensemble mean surface temperature we use the area-weighted root mean square error, as in Bilbao et al. (2024). To determine the impact of the volcanic forcings on the quality of the predictions we also compute for each grid cell whether the observations are outside the 95% range of the ensemble members by computing the 5th and 95th percentiles.



| Experiment | Description |
|---|---|
| DCPP-hindcast-A | 10 member ensemble simulations of 10 forecast years initialised in November. It uses prescribed CMIP6 forcings including the stratospheric aerosol forcing. |
| DCPP-hindcast-C | Repeat the s1962, s1981 and s1990 hindcasts but replacing the stratospheric aerosol forcing with the mean of 1850-2015, equivalent to no volcanic forcing. |
| DCPP-EVA | Repeat the s1962, s1981 and s1990 hindcasts but with the EVA volcanic forcing. |
| DCPP-EVA_H | As DCPP-EVA but with the EVA_H volcanic forcing. |
| DCPP-EVA_H_Carn16 | As DCPP-EVA_H s1990 but with the EVA_H volcanic forcing including all the eruption from Carn et al. (2016). |

**Table 1.** EC-Earth3 decadal prediction experiments.

## 3 Results

### 3.1 Volcanic Forcings

We start by comparing the global mean AOD at 530 nm for the volcanic forcings produced with EVA and EVA_H including the 1963 eruption of Mount Agung, the 1982 eruption of El Chichón and the 1991 eruption of Mount Pinatubo, with the CMIP6 stratospheric aerosol forcing (Luo, 2018) (Figure 1). The CMIP6 forcing may be considered a best estimate, although the AOD after the eruptions of Mount Agung and El Chichón are very uncertain due to the limited amount of observations. Overall the three forcings show a comparable global mean AOD (at 530 nm) evolution following the three volcanic eruptions, however, there are some evident differences among them. These are more pronounced for the eruptions of Agung and El Chichón. For these eruptions, the peak AOD simulated by both EVA and EVA_H is smaller than in CMIP6 by ∼20% and ∼40%, respectively. As for the temporal evolution, the EVA and EVA_H forcings compare reasonably well with CMIP6 for the eruption of Agung, but decay faster for El Chichón. In the case of the eruption of Pinatubo, both EVA and EVA_H forcings compare better with the CMIP6 values, however some differences remain. For the EVA forcing, the peak AOD is very similar in magnitude to CMIP6, but the forcing rises faster and decays sooner. In contrast, the decay in the EVA_H forcing compares better to CMIP6, but its peak AOD is smaller by approximately 14%. Our EVA and EVA_H forcing time series only includes the three main eruptions, which dominated the changes in AOD, but other smaller eruptions have occurred. Consequently, we produce another forcing time series with EVA_H using the complete Carn et al. (2016) emission inventory, referred to as EVA_H_Carn16. This suggests that the apparent underestimation of the Pinatubo forcing (∼18 Tg $SO_2$ in Carn et al., 2016) is mostly resolved when including all eruptions, most importantly the 1991 eruption of Cerro Hudson (∼4 Tg $SO_2$ in Carn et al., 2016).

Figure 2 shows that both EVA and EVA_H forcings also have some discrepancies with CMIP6 in terms of the latitudinal structure of the forcing. The CMIP6 forcing indicates that the eruption of Agung had a stronger aerosol forcing in the Southern Hemisphere, the eruption of El Chichón produced a stronger forcing in the Northern Hemisphere and the eruption of Pinatubo produced a mostly hemispherically symmetric forcing (Figure 2a-c). For the eruption of Pinatubo, EVA simulates a forcing with stronger maxima in the equator and a weaker forcing in the high latitudes with respect to CMIP6 (Figure 2f). On the other hand, EVA_H simulates comparatively well the three maxima in the CMIP6 forcing (in the equator and Northern and Southern Hemispheres), however the magnitude of the forcing is overall underestimated, especially in the tropics (Figure 2i). Overall, EVA simulates the magnitude of the maxima in the equator closer to CMIP6, but the overall latitudinal structure is better captured by EVA_H. For the other two eruptions, which are more uncertain, EVA_H simulates a forcing structure similar




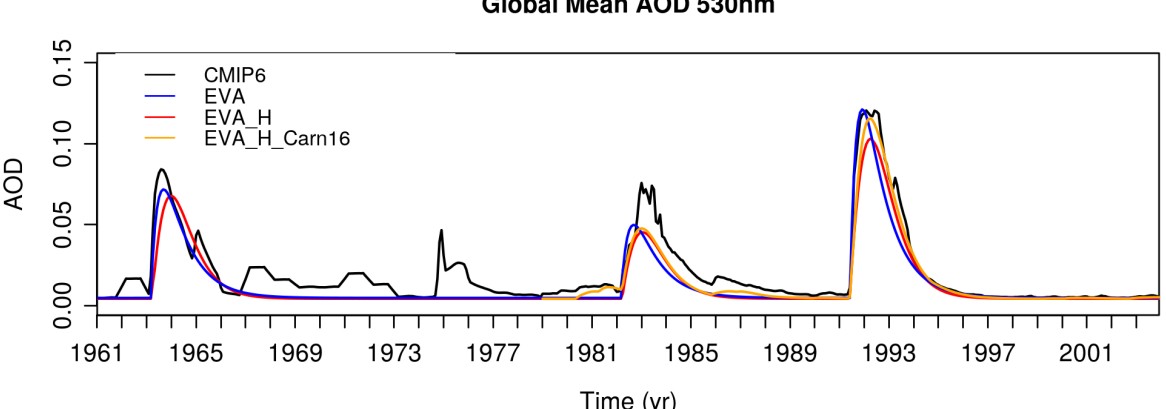

**Figure 1.** Global mean stratospheric aerosol optical depth at 530 nm. The EVA and EVA_H forcings only have data for the eruptions of Mount Agung in 1963, El Chichón in 1982 and Mount Pinatubo in 1991. The EVA_H_Carn16 forcing was generated with the data from Carn et al. (2016) which includes all volcanic eruptions from 1978-2004.

to the one for Pinatubo, failing to account for the latitudinal asymmetry of the forcing (Figure 2g and h). This is not the case for EVA, for which the latitudinal asymmetry of the forcing of Mount Agung and El Chichón are partly captured (Figure 2d and e), thanks to the use of hemispheric asymmetry factor, not accounted for by EVA_H. However the AOD values are weaker
by ∼40% in the Southern Hemisphere for the eruption of Agung and by ∼70% in the Northern Hemisphere for the eruption of El Chichón, with respect to CMIP6.

Figure 2 shows that both EVA and EVA_H forcings also have some discrepancies with CMIP6 in terms of the latitudinal structure of the forcing. The CMIP6 forcing indicates that the eruption of Agung had a stronger aerosol forcing in the Southern Hemisphere, the eruption of El Chichón produced a stronger forcing in the Northern Hemisphere and the eruption of Pinatubo
produced a mostly hemispherically symmetric forcing (figure 2a-c). For the eruptions of Agung and El Chichón, whose forcings are more uncertain than Pinatubo, EVA partly captures the latitudinal asymmetry of the forcing (figure 2d and e), thanks to the use of the hemispheric asymmetry factor. However, the AOD values are weaker by ∼40% in the Southern Hemisphere for the eruption of Agung and by ∼70% in the Northern Hemisphere for the eruption of El Chichón, with respect to CMIP6. EVA_H, in contrast, simulates a forcing structure similar to the one for Pinatubo (figure 2i), failing to account for the latitudinal asymmetry
(figure 2g and h). The forcings magnitudes are also considerably weaker in EVA_H with respect to CMIP6. For the eruption of Pinatubo, EVA simulates a forcing with stronger maxima in the equator and a weaker forcing in the high latitudes with respect to CMIP6 (figure 2f). On the other hand, EVA_H simulates comparatively well the three maxima in the CMIP6 forcing (in the equator and Northern and Southern Hemispheres), however the magnitude of the forcing is overall underestimated, especially in the tropics (figure 2i). Overall, EVA simulates the magnitude of the maxima in the equator closer to CMIP6, but the overall
latitudinal structure is better captured by EVA_H.





Comparing the EVA_H and EVA_H_Carn16 forcings (Figure S1) we find that including the eruption of Cerro Hudson produces a stronger forcing in the Southern Hemisphere, which increases the similarity with the CMIP6 forcing. This suggests a potential role of the eruption of Cerro Hudson in the latitudinal distribution in 1991, making Pinatubo seem more hemispherically symmetric than it was. Note also that there is a minor increase in AOD in the EVA_H-Carn16 forcing in the Northern
hemisphere, which is due to other small Northern Hemisphere eruptions included in the Carn et al. (2016).

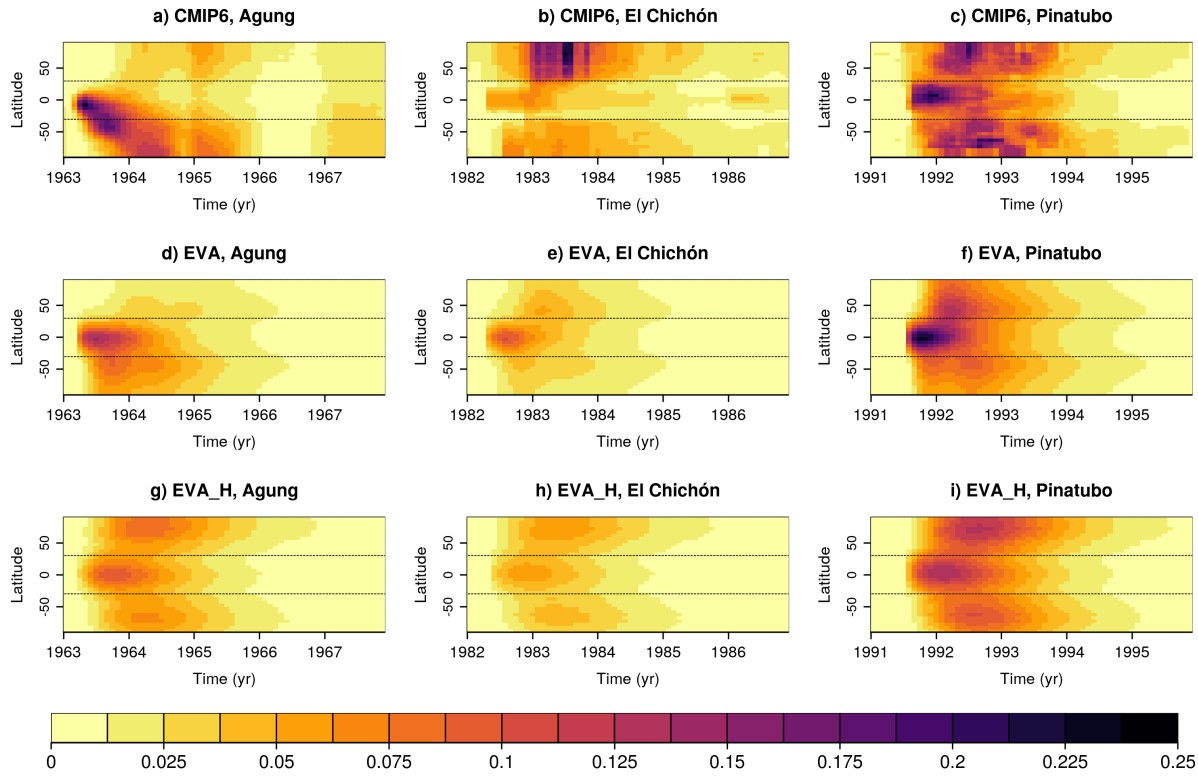

**Figure 2.** Stratospheric aerosol optical depth at 530 nm as a function of time and latitude for the eruptions of Mount Agung in 1963 (first column), El Chichón in 1982 (second column) and Mount Pinatubo in 1991 (third column) for CMIP6, EVA and EVA_H.

Figure 3 shows the tropical mean extinction as a function of time and altitude. For CMIP6, the altitude of peak tropical extinction clearly decreases with time for all three eruptions considered, from ∼24 km (1963) to ∼20 km (1966) for Agung, ∼22 km (1983) to ∼18 km (1986) for El Chichon and ∼22 km (1991) to ∼19 km (1994) for Pinatubo. The forcing vertical structure is independent of time for the eruptions considered in EVA, with a peak tropical extinction at 25 km. In contrast, the
developments in EVA_H allow temporal and eruption-dependent peak extinction altitude, which is more realistic. While for the Agung forcing produced with EVA_H the peak extinction does not vary, for El Chichon it decreases from ∼23 km (1983) to ∼20 km (1986) for El Chichon and ∼23.5 km (1991) to ∼19.5 km (1994) for Pinatubo.





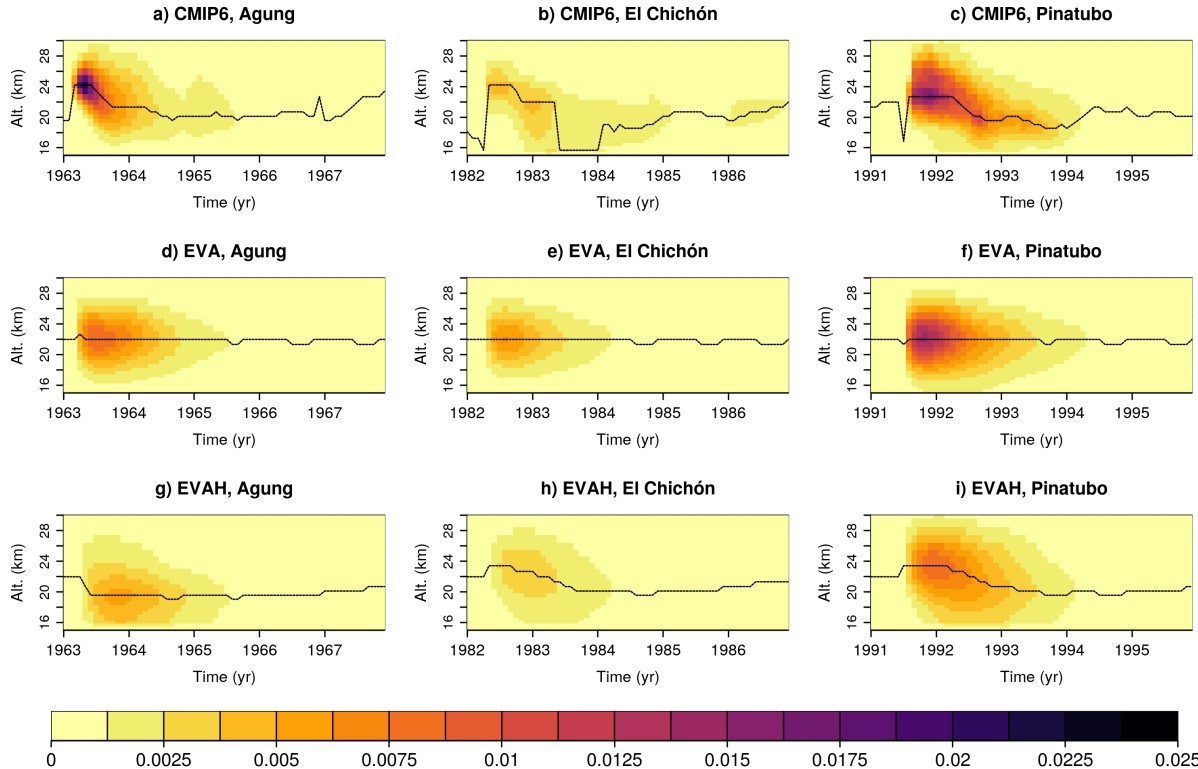

**Figure 3.** Tropical-mean (30°N-30°S) extinction (km$^{-1}$) at 530 nm as a function of time and altitude for the eruptions of Mount Agung in 1963 (first column), El Chichón in 1982 (second column) and Mount Pinatubo in 1991 (third column) for CMIP6, EVA and EVA_H. Logarithmic colorbar scale. The dashed line indicates the height of the maximum AOD.

## 3.2 Climate response to the volcanic forcings

### 3.2.1 Global mean climate response

We first analyse the global mean net top-of-atmosphere (TOA) radiation flux response (i.e., hindcasts with volcanic forcing minus hindcasts with no volcanic forcing), calculated as anomalies of incoming shortwave minus outgoing shortwave and out-going longwave radiation. Figure 4a-c shows a post-volcanic decrease in global mean TOA radiation flux response which is qualitatively similar in the three hindcast experiments (DCPP-A, DCPP-EVA and DCPP-EVA_H). The ensemble mean negative TOA flux anomalies are statistically significant following the eruptions of Mount Agung and Pinatubo, which last for a year and two years respectively, while for the eruption of El Chichón shows a weaker and barely significant response (c.f. Bilbao et al., 2024). Comparing the ensemble mean TOA flux responses in the hindcast sets for each eruption we find some notable differences in the magnitude and the temporal evolution, which strongly relate to the volcanic forcings used in each. DCPP-EVA and in particular DCPP-EVA_H show an overall weaker response compared to DCPP-A following the eruptions





of Mount Agung and El Chichón. For the latter eruption, despite being weak and therefore harder to detect robust signals, DCPP-EVA shows an initial strong response and then in both DCPP-EVA and DCPP-EVA_H the anomalies recover earlier than in DCPP-A, since the CMIP6 forcing is overall stronger (Figure 1). For the eruption of Mount Pinatubo, the magnitude of the response in DCPP-EVA compares well with DCPP-A and may even be slightly stronger, while in DCPP-EVA_H the response is considerably weaker. There are also some differences in the temporal evolution in DCPP-EVA, as the EVA forcing decays earlier than in EVA_H and CMIP6.

The global mean surface temperature responses relate well to the global mean TOA fluxes which show similar results but the differences become more apparent (Figure 4d-f). For the eruptions of Agung and El Chichón, where the magnitude of both EVA and EVA_H forcings is weaker than in CMP6, the post-volcanic cooling in DCPP-EVA and DCPP-EVA_H is consequently weaker with respect to DCPP-A (Figure 4d and e). The DCPP-EVA and DCPP-EVA_H ensemble mean responses to Agung are only statistically significant the second and third years following the eruption with no significant response found for the eruption of El Chichón. For the eruption of Pinatubo the negative global surface temperature responses are greater than for the other eruptions and remain statistically significant for up to six years (Figure 4f). Consistent with the TOA flux, in DCPP-EVA, the negative global surface temperature anomalies tend to be greater than DCPP-A, especially in the first year following the eruption, while in DCPP-EVA_H the response is weaker than DCPP-A (Figure 4f).

The global mean temperature in the lower stratosphere (50 hPa) shows strong post eruption warming anomalies with smaller ensemble spread in comparison to other variables (Figure 4g-i), reinforcing the fundamental differences induced by the forcings. In the DCPP-EVA hindcasts the lower stratospheric anomalies show particular differences with respect to DCPP-A depending on the eruption. For the eruption of Agung the temporal evolution of the warming is consistent with DCPP-A but slightly weaker in magnitude, for El Chichón the warming happens earlier and is slightly stronger and for Pinatubo the magnitude is stronger but decays earlier. In contrast, in DCPP-EVA_H the lower stratospheric anomalies are consistently weaker than in DCPP-A and DCPP-EVA_H for the three eruptions, although there are some similarities in the temporal evolution.

For the eruption of Mount Pinatubo, the weaker TOA flux response in DCPP-EVA_H is, at least partly, due to omitting the eruption of Cerro Hudson. As indicated by the EVA_H_Hud experiment, when this eruption is also included, the ensemble mean TOA radiation flux response is still weaker compared to DCPP-A (by ∼18%) (Figure 4c), reducing the difference with CMIP6 by half. Consequently, the global mean surface temperature response magnitude and temporal evolution agreement with DCPP-A improves, and is closer than the other two hindcasts (DCPP-EVA and DCPP-EVA_H). In contrast, for the global mean temperature response in the lower stratosphere (50 hPa), including the eruption of Cerro Hudson in the EVA_H forcing marginally increases the global mean temperature response in the lower stratosphere.

### 3.2.2 Spatiotemporal Characteristics of the Radiative Response

Next, we examine how latitudinal differences in the forcings influence the geographical patterns of the response. Figure 5 shows the zonal mean TOA flux response in the first year following the eruptions, when the TOA response is at its peak. Consistent with the volcanic forcings (Figure 2), the largest TOA flux decrease—statistically significant in all hindcast experiments—occurs in the tropics. However, there are noticeable latitudinal differences in the forcings for the three eruptions, which

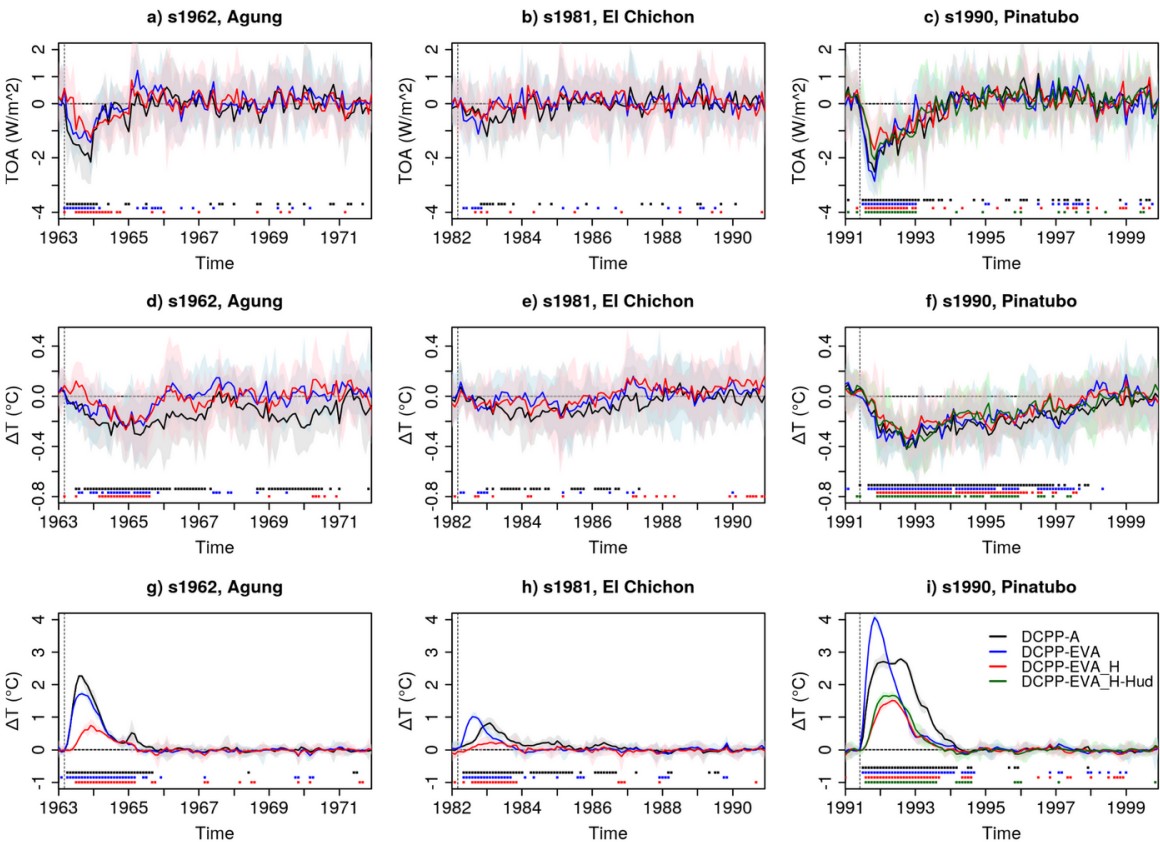

**Figure 4.** Global mean (a-c) top-of-atmosphere net radiative flux (Wm$^{-2}$), (d-f) surface air temperature(°C) (d-f) and (g-i) lower stratospheric (50hPa) temperature (°C) responses to the volcanic eruptions (volc - no volc), for DCPP-A (CMIP6) in black, DCPP-EVA in blue, DCPP-EVA_H in red and DCPP-EVA_H_Hud in green. The shading is the multi-model member spread calculated as the 10th and 90th percentiles of the ensemble. Filled squares at the bottom part of the figure indicate statistically significant differences according to a bootstrap with resampling with 1000 iterations.





reflect variations in volcanic forcing. Most notably, neither DCPP-EVA nor DCPP-EVA-H reproduce the TOA flux response in the Southern and Northern Hemispheres after the eruptions of Mount Agung and El Chichón, respectively (Figure 5a and b).

Nevertheless, for Mount Agung, DCPP-EVA shows a statistically significant TOA flux decrease in the Southern Hemisphere, as the EVA volcanic forcing captures, at least partly, the latitudinal structure of the forcing through the hemispheric asymmetry parameter, though the effect is much weaker than in the CMIP6 forcing. For the Pinatubo eruption, which is nearly hemispherically symmetric, the negative TOA flux response in DCPP-EVA is comparable to DCPP-A but slightly stronger at the equator, with the peak extending further south (Figure 5c). In contrast, DCPP-EVA-H shows a considerably weaker negative TOA flux

response along the equator (around -2 $Wm^{-2}$).

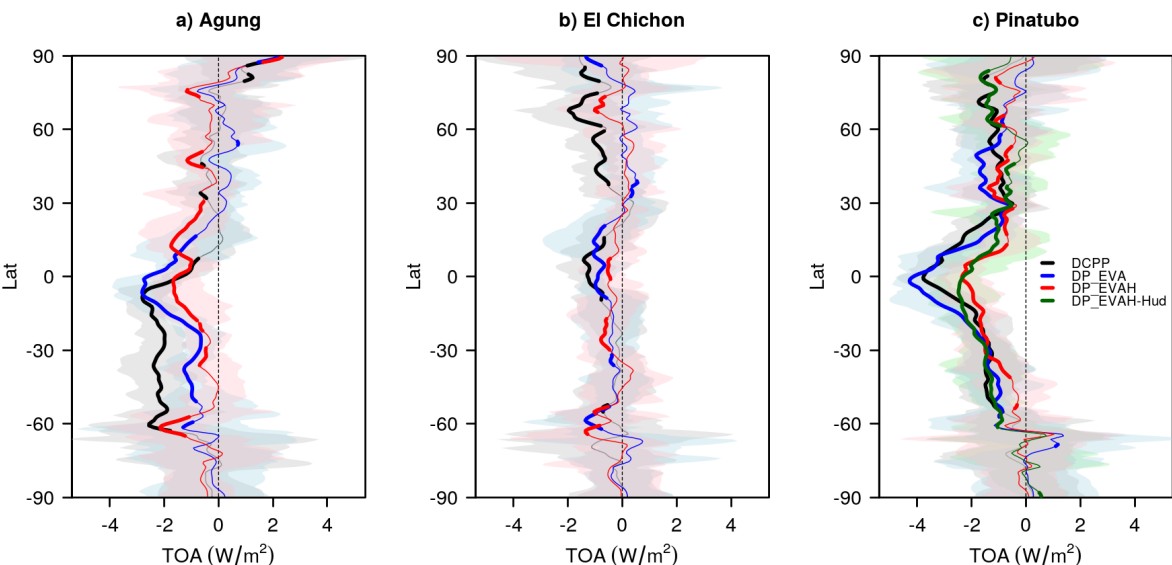

**Figure 5.** Zonal mean TOA radiation flux ($Wm^{-2}$) response (volc - no volc) the first year following the eruptions (June-May) in DCPP (CMIP6), DCPP-EVA and DCPP-EVA_H. The ensemble mean for each hindcast experiment is shown. The shading is the member spread calculated as the 10th and 90th percentiles of the ensemble. Thicker lines indicate statistically significant anomalies according to a bootstrap with resampling with 1000 iterations (see the Methods section)

Latitudinal differences in the TOA flux response influence the surface temperature response, which can last several years, depending on the eruption's magnitude. In the first year following the eruptions, which roughly coincides with the period of post volcanic global mean surface cooling, the surface temperature response patterns are characterised by cooling over the tropics and subtropics (Figure 6). This cooling is generally consistent across the hindcast experiments and volcanic eruptions,

despite some noise due to the small ensemble size. However, notable differences align with variations in the TOA radiation flux response. For example, both DCPP-EVA and DCPP-EVA_H simulate weaker tropical cooling following the eruption of Mount Agung, with the effect being more pronounced in DCPP-EVA_H. Similarly, the cooling response is also weaker in




DCPP-EVA_H after the eruption of Mount Pinatubo. For the DCPP-EVA_H-Hud predictions the cooling response is similar, with slightly stronger cooling due to adding the contribution of Cerro Hudson to the volcanic forcing (Figure S2a).

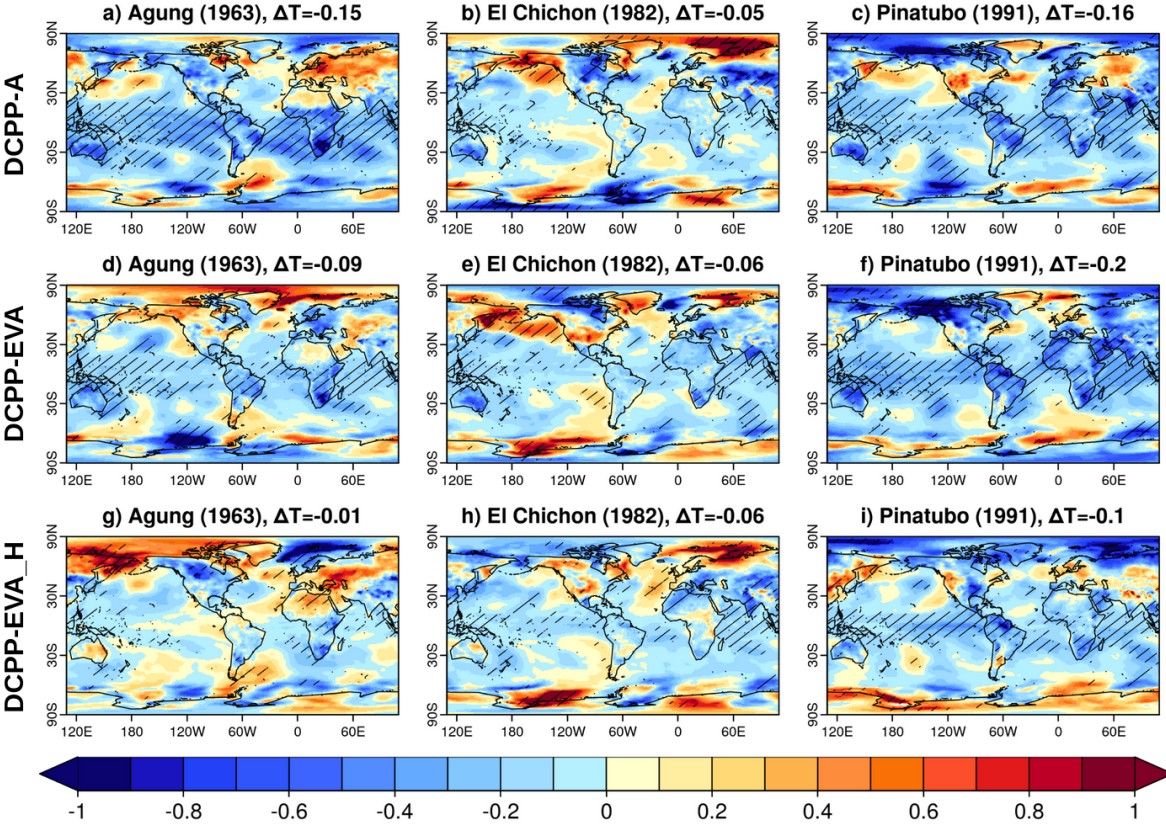

**Figure 6.** Ensemble mean surface air temperature (°C) response (volc - no volc) for the first year following the eruptions (June-May) in DCPP (CMIP6), DCPP-EVA and DCPP-EVA_H. Hatching indicates statistically significant anomalies according to a bootstrap with resampling with 1000 iterations. The titles include the global mean surface temperature response.

For years 2 to 5 after the eruptions, as the global mean surface temperature recovers, cooling spreads worldwide, with the strongest effects seen in the tropics, subtropics, and Arctic, depending on the eruption's magnitude (Figure 7). For this period, the volcanic impact on surface temperature becomes clearer due to longer temporal averaging, revealing significant differences across the hindcasts. Notably, the DCPP-EVA and DCPP-EVA_H hindcasts show much weaker volcanic cooling compared to DCPP-A for the Mount Agung and El Chichón eruptions (Figure 7d, e, g and h). In contrast, for the Pinatubo eruption, these hindcasts simulate widespread cooling, with the strongest cooling over the Arctic (Figure 7f, i). However, there are some differences from DCPP-A, attributable to the volcanic forcing differences described previously: DCPP-EVA shows slightly stronger tropical cooling than DCPP-A, while DCPP-EVA_H simulates generally weaker anomalies overall. For the DCPP-EVA_H-Hud hindcasts the response patterns are similar, again with a slightly stronger cooling response (Figure S2b).





In forecast years 6-9 (Figure S3), surface temperature anomalies generally recover and weaken globally, except in the DCPP-A experiment for the Mount Agung eruption. However, this cooling is most likely not caused by the Agung eruption itself but by other smaller volcanic eruptions included in the CMIP6 volcanic forcing. Notably, on these forecast times, the Arctic is a region where cool anomalies persist in most hindcast experiments, particularly in response to the Mount Pinatubo eruption.

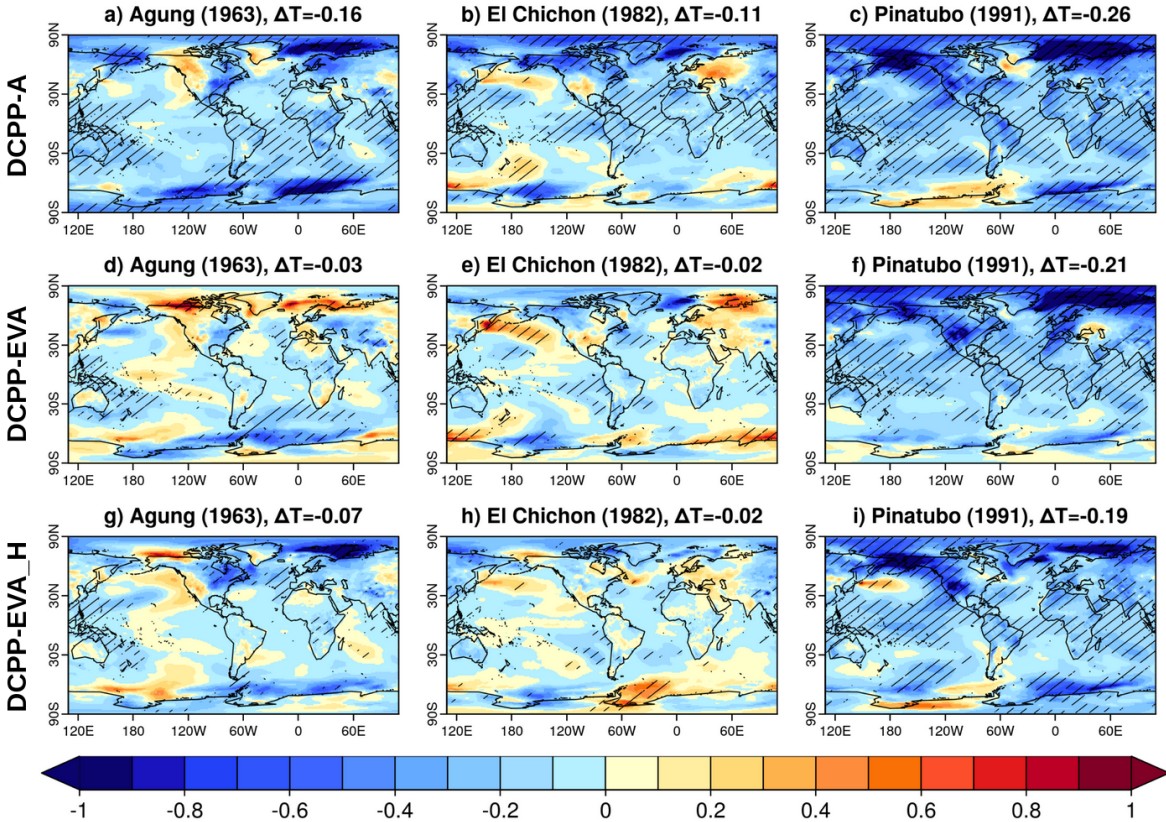

**Figure 7.** Ensemble mean surface air temperature (°C) response (volc - no volc) for years 2-5 following the eruptions (June-May) in DCPP (CMIP6), DCPP-EVA and DCPP-EVA_H. Hatching indicates statistically significant anomalies according to a bootstrap with resampling with 1000 iterations. The titles include the global mean surface temperature response.

### 3.2.3 Comparison of the predicted response with observations

To assess the potential impact of using forcings generated by EVA and EVA_H in operational forecasts, we compare the predicted anomalies from the hindcast experiments with the ERA5 reanalysis and observational datasets available.

Figures 8a-c show the global mean TOA flux anomalies. For the eruptions of Mount Agung and El Chichón, the hindcast experiments consistently simulate weaker global mean TOA fluxes compared to the ERA5 data. Additionally, both the DCPP-EVA and DCPP-EVA-H models produce weaker anomalies than the DCPP model. However, these results should be interpreted





cautiously due to low confidence in pre-satellite TOA flux observations. For the Mount Pinatubo eruption, the global mean
TOA flux anomalies can also be compared against the Deep-C version 5 reconstruction (Liu et al., 2020), which is based on
CERES satellite observations. For this eruption, all hindcast experiments, except DCPP-C, show a strong alignment with both
ERA5 and Deep-C, underscoring the significance of including volcanic forcing to accurately simulate the post-eruption TOA
flux decrease. This agreement suggests that the predicted response in global mean flux is realistic. While the predicted TOA
flux anomalies generally align well with observational data, the strongest agreement is observed in the DCPP and DCPP-EVA
simulations. In contrast, the DCPP-EVA-H and DCPP-EVA_H_Hud simulations show slightly weaker anomalies, consistent
with previous findings.

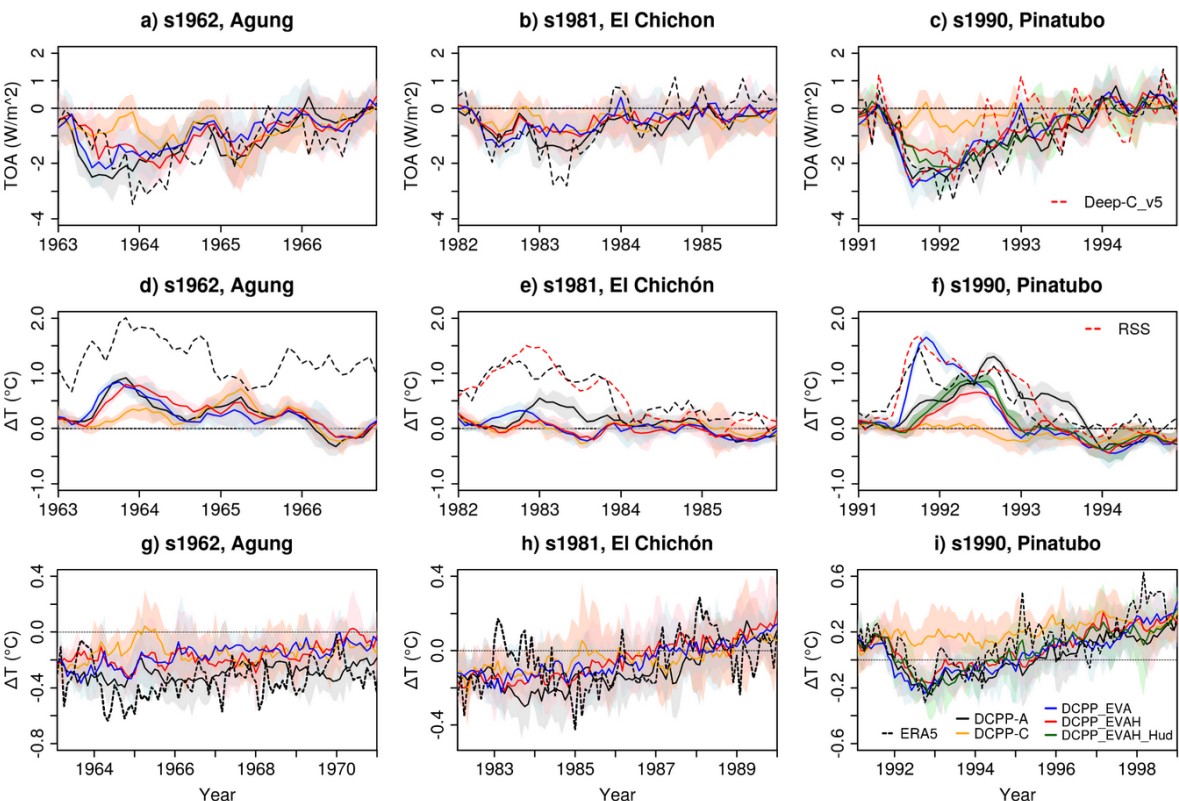

**Figure 8.** Global mean (a-c) top-of-atmosphere net radiative flux (Wm$^{-2}$), (d-f) lower stratospheric (100hPa) temperature (°C) and (g-i)
surface air temperature (°C) anomalies for the decadal hindcasts and observational products. The anomalies for the TOA flux and lower
stratospheric temperature have been computed with respect to the period 1985-2015, while for surface temperature the reference period is
1970-2005 (see methods for further information). For the lower stratospheric temperature anomalies the global mean was computed from
83°S and 83°N as the RSS observational dataset is limited to this range. The ensemble mean for each hindcast is shown. The shading is the
multi-model member spread calculated as the 10th and 90th percentiles of the ensemble.





Comparing the latitudinal average of the predicted TOA flux anomalies for the Mount Pinatubo eruption with the Deep-C version 5 dataset shows that the DCPP and DCPP-EVA models match the observed values well, especially the negative anomalies in the tropics (Figures 9). The observations tend to be within the ensemble spread of the predictions, although in
DCPP-EVA the anomalies may be slightly overestimated in the tropics. In contrast, the DCPP-EVA_H and DCPP-EVA_H_Hud hindcasts substantially underestimate the negative anomalies in the tropics during the first six months following the eruption, although they show similar values to DCPP-A and DCPP-EVA in later months. It is also worth noting that all the predictions struggle to reproduce the changes in the high latitudes, but this might be unrelated to the volcanic forcing.

Figures 8d-f show the global mean temperature anomalies at 100 hPa for the prediction experiments, ERA5, and obser-
vational data from RSS. For the Mount Agung eruption, where direct observations are unavailable, all hindcast experiments except DCPP-C are similar, though consistently lower than ERA5 throughout the period, suggesting that comparisons should be approached with caution. For the El Chichón and Mount Pinatubo eruptions, observations from RSS are available and as mentioned in Methods, the comparison with RSS must be approached with caution. However it is reassuring that ERA5 aligns closely with this dataset. In the case of El Chichón, the predicted global mean lower stratospheric temperature anomalies are
weaker than observed. For Mount Pinatubo, DCPP-EVA initially shows the closest agreement with RSS and ERA5; however, the temperature anomalies decrease more rapidly than observed, consistent with prior findings. The DCPP and DCPP-EVA_H hindcasts both underestimate the initial warming, but DCPP shows better alignment in later years while DCPP-EVA_H decays too soon. Finally, DCPP-C fails to simulate lower stratospheric warming, underscoring the importance of volcanic forcing in capturing the post-eruption response accurately.

Figure 8g-i shows the predicted global mean surface temperature anomalies alongside ERA5. Since surface temperature observations are more reliable, these comparisons should provide robust insights. Overall, the EC-Earth3 hindcasts that include volcanic forcing (DCPP-A, DCPP-EVA, and DCPP-EVA_H) align more closely with ERA5 global mean surface temperature anomalies than the hindcasts without volcanic forcing (DCPP-C). Consistent with the multi-model analysis of (Bilbao et al., 2024), while the volcanic impact is less evident for the Mount Agung and El Chichón eruptions—likely due to greater
observational uncertainty and their moderate magnitudes—volcanic forcing is particularly crucial for accurately reproducing the observed global temperature cooling following the Pinatubo eruption in the early 1990s. For Pinatubo, despite differences between the CMIP6, EVA, and EVA_H forcings in the hindcasts, the observed anomalies generally fall within the uncertainty range of the hindcast ensembles (between the 10th and 90th percentiles). It is worth noting that, consistent with previous findings, the global mean surface temperature response in DCPP-EVA initially shows stronger cooling than observed, while
DCPP-EVA_H simulates slightly weaker peak cooling.

Since the volcanic forcing significantly improves the global mean surface temperature predictions—particularly for the Pinatubo eruption—we also analyse regional anomalies to assess the potential improvements in the response patterns. Figure 10 shows the surface temperature anomalies in ERA5 and the ensemble mean predictions for forecast years 1–5, the period when the volcanic signal is strongest. The ensemble mean surface temperature anomaly patterns are generally smoother with
respect to ERA5, indicating the limitations of decadal forecast systems in predicting regional variations on these timescales. Across the experiments, the predicted anomaly patterns are largely similar, except for the DCPP-C hindcasts. These hindcasts



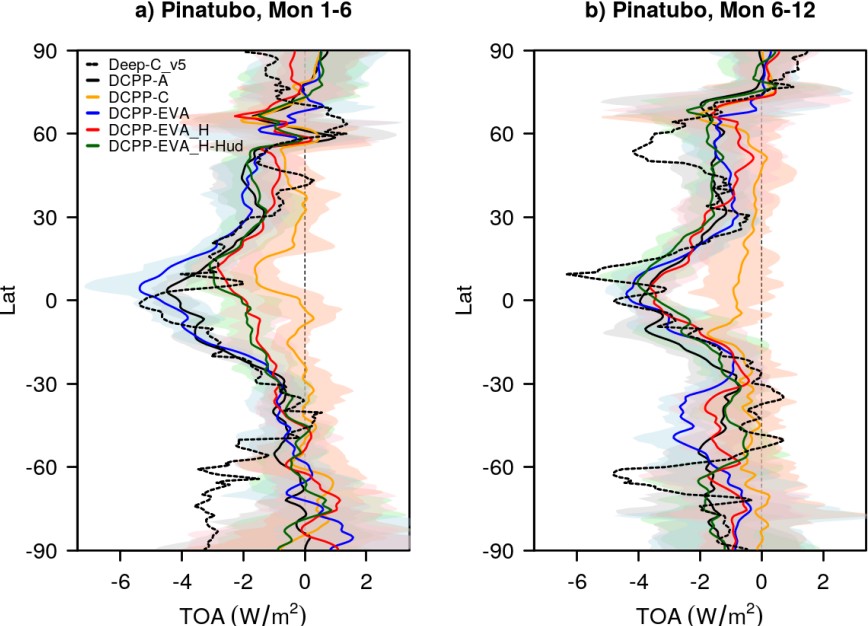

**Figure 9.** Zonal mean TOA radiation flux anomaly (Wm$^{-2}$) timeseries for (a) months 1-6 and (b) 7-12 following the eruption of Mount Pinatubo. The anomalies are computed with respect to the period 1985-2015. The shading is the multi-model member spread calculated as the 10th and 90th percentiles of the ensemble.

simulate warmer anomalies (as they omit volcanic forcing) leading to larger forecast errors (except for Mount Agung in DCPP-EVA), as indicated by the area-weighted RMSE, though the differences are small. Additionally, this experiment shows a higher proportion of grid points where ERA5 falls outside the 95% ensemble range. These discrepancies are most prominent in regions

where volcanic impacts are strongest, particularly in the tropics, as shown in figure 7. Interestingly, in some regions, the DCPP-C hindcasts may appear to improve consistency with observations, such as in the tropical Pacific. However, this improvement probably arises for the wrong reasons. For instance, in the case of the Pinatubo eruption, the warmer conditions in the tropical Pacific in DCPP-C are not due to a better prediction of the El Niño conditions but rather the absence of the volcanic radiative response. Overall, the volcanic forcing enhances prediction accuracy, despite the small ensemble size.

Comparing the hindcast experiments with volcanic forcings (DCPP-A with the DCPP-EVA, and DCPP-EVA_H), we find that the area-weighted RMSE is similar among the volcanic eruptions. Additionally, including the eruption of Cerro Hudson make little difference in this respect as shown in Figure S4 for the DCPP-EVA_H-Hud experiment. The fact that none of the experiments outperform the others could be because the forecast errors unrelated to volcanic forcing (i.e. internal variations) are larger than those attributable to differences in volcanic forcing. In terms of the percentage of global area where ERA5

observations fall outside the 95% range of the ensemble members, we find that all experiments coincide in the regions for each



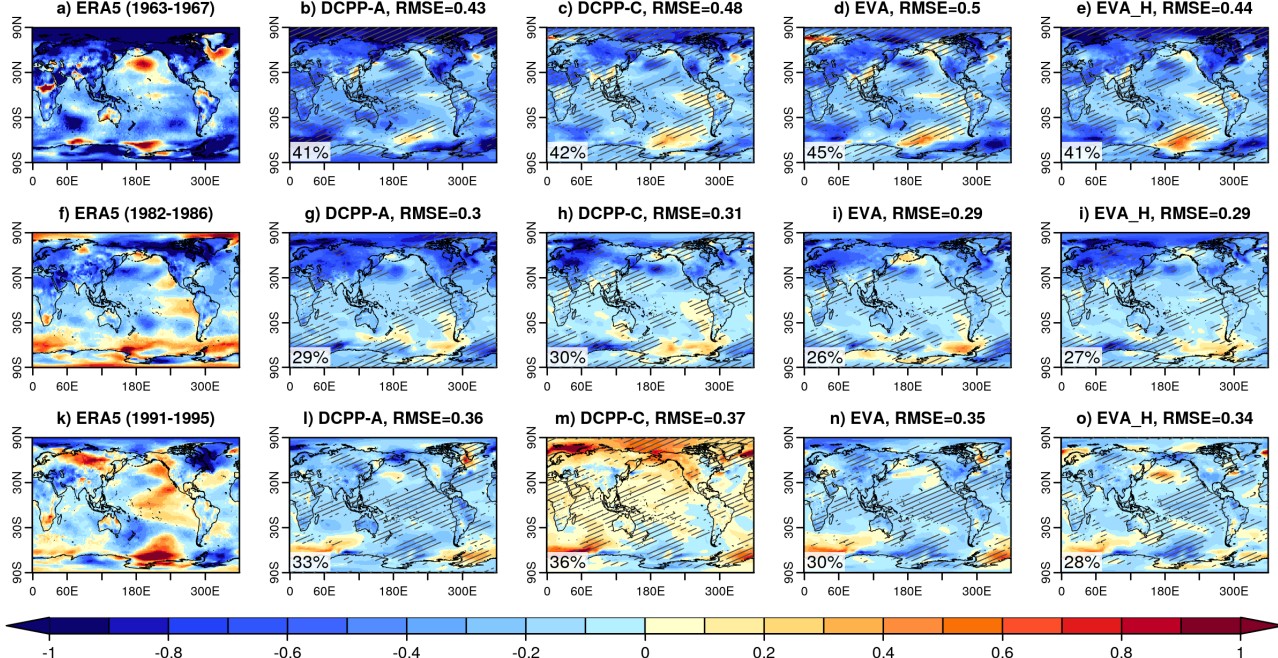

**Figure 10.** Surface air temperature anomalies (°C) for forecast years 2-5 following the eruptions (years are defined from Jan-Dec) in ERA5 and for the predictions ensemble mean initialised in 1962, 1981 and 1990 for the DCPP-A, DCPP-C, DCPP-EVA and DCPP-EVA_H hindcasts. The anomalies have been computed with respect to the period 1971-2015 (see methods). Stippling indicates where the observations fall outside of the 90% range of the ensemble and the percentage at the bottom left corner of each map indicates the percentage of grid-points outside this range.

of the eruptions, with small differences. This indicates that using the forcings from EVA or EVA_H provide reasonable results and could be used in operational predictions, particularly for Pinatubo-like eruptions.

## 4 Discussion and Conclusions

Due to the unpredictable nature of volcanic eruptions, if one occurs, operational decadal forecasts will no longer be accurate.
Given that large volcanic eruptions have climate impacts, for a rapid response, decadal hindcasts could be rerun with volcanic forcings generated with EVA and EVA_H once the eruption characteristics are known (Sospedra-Alfonso et al., 2024). While these tools have been widely used to carry out sensitivity studies to volcanic forcings, their use in real time forecasts is largely unexplored. In this paper we have compared the volcanic forcings generated with EVA and EVA_H for the last three major volcanic eruptions with the CMIP6 volcanic forcing, and evaluated their climate response in decadal predictions with the BSC
decadal forecast system following a similar protocol to DCPP component C (Boer et al., 2016).



We have compared the volcanic forcings for the eruptions of Agung, El Chichón and Pinatubo, generated with EVA and EVA_H with the CMIP6 forcing, and quantified their magnitude and latitudinal structure differences. Although EVA_H exhibits a more similar temporal and vertical evolution to CMIP6 than EVA, it fails to capture the hemispherical asymmetry for tropical eruptions like Agung and El Chichón owing to the absence of EVA-like parameterization enabling preferential transport to one hemisphere.

The differences in the volcanic forcings lead to variations in the radiative responses simulated by EC-Earth3 decadal predictions, as shown by the differences in TOA fluxes, surface temperature and lower stratospheric temperatures. For the eruptions of Mount Agung and El Chichón, both hindcasts with EVA and EVA_H forcings produce weaker temperature anomalies at the surface and lower stratosphere compared to DCPP-A. As a result of the latitudinal structure of the forcings, the surface temperature patterns in the hindcasts with EVA and EVA_H show very weak responses in the Southern Hemisphere following Agung and in the Northern Hemisphere for El Chichón, in comparison to DCPP-A. For the eruption of Pinatubo the hindcasts with the EVA forcing exhibit a stronger radiative response than DCPP-A, while in the hindcasts with the EVA_H forcing it is weaker as it was calibrated accounting for the Cerro Hudson 1991 eruption. Despite including this eruption in the EVA_H volcanic forcing the effect is small, although it contributes to strengthening the anomalies. The surface temperature patterns show that the main differences are shown in the tropics, where the EVA_H show a much weaker response with respect to the hindcasts with EVA forcing and DCPP-A.

We have also compared the predicted anomalies for the three hindcasts with the available observations for TOA fluxes, lower stratospheric temperature and surface temperature. These results highlight the importance of including the volcanic forcing to make skillful climate predictions following major volcanic eruptions, particularly in capturing the radiative effects, which are largely consistent in decadal prediction systems (e.g. Bilbao et al., 2024). Despite this relevant result, the hindcasts show limitations in predicting the observations. For the eruptions of Mount Agung and El Chichón, EVA and EVA_H may exhibit some limitations in reproducing the magnitude and in particular the latitudinal structure of the volcanic forcing. However these eruptions are subject to large observational uncertainties, in terms of direct AOD and emissions as well as the TOA flux and atmospheric temperature observations over that period. This is evident from the evaluation of the DCPP-A hindcasts, which despite having been run with the CMIP6 forcing (one of the best estimates currently available), show large errors. In contrast, for the eruption of Mount Pinatubo, the hindcasts predict the observed post-volcanic effect more accurately. Although the hindcasts with EVA forcings weakly overestimate the radiative response while the hindcasts with EVA_H forcings underestimate it. Overall, our study suggests that despite these deficiencies, both EVA and EVA_H forcings can be reasonable choices for predicting the post-volcanic radiative effects, given the inherent forecast uncertainty.

Previous studies have shown that volcanic eruptions can also lead to atmospheric and oceanic dynamical changes (see Marshall et al. (2022) for a review). These dynamical changes, however, have larger uncertainties with respect to the radiative impacts, as large ensembles are required to detect responses, can be model dependent (e.g. Bilbao et al., 2024), can be affected by the background climate conditions (e.g. Zanchettin et al., 2022) and there is evidence that models might be deficient in simulating some of the impacts (e.g. Wu et al., 2023). Therefore, a caveat of this study is that due to the small ensemble size of these experiments it has not been possible to detect and evaluate the dynamical impacts. Previous studies have shown that the



post eruption tropical stratospheric warming impacts the atmospheric circulation by strengthening the polar vortex, leading to positive NAO-like conditions, which may result in warming of the North Eurasian continent the first winter after the eruption (e.g. Hermanson et al., 2020). Here we find a significant response in the stratospheric temperatures (Figure 4g-i), but we do not detect an acceleration of the Northern Hemisphere stratospheric polar vortex, possibly due to the strong variability and the

small ensemble size. Furthermore, models have been shown to be affected by the signal to noise problem of the atmospheric circulation, especially in the North Atlantic, whereby they might be underestimating the real response (Scaife and Smith, 2018; Smith et al., 2020; Hermanson et al., 2020). Other dynamical impacts include changes in the North Atlantic Ocean, which may lead to an acceleration of the Atlantic Meridional Overturning Circulation, and El Niño Southern Oscillation (ENSO) (e.g. Hermanson et al., 2020; Bilbao et al., 2024). In these simulations it has not been possible to detect such responses, again

probably due to the small ensemble size and also because EC-Earth3 is one of the models with weak response in this respect as shown in Bilbao et al. (2024).

The results of this study provide further insights on the effects of volcanic eruptions on climate and their predictability, in particular on the expected uncertainty of using the EVA and EVA_H forcings in operational decadal predictions, as well as informing the future development of these tools. However, we are limited to a small ensemble and one prediction system.

To make further progress we suggest other modelling centres to carry out similar predictions with EVA and EVA_H forcings for past volcanic eruptions. While we expect the radiative effects to be consistent, based on the results of recent papers (e.g. Zanchettin et al., 2022; Bilbao et al., 2024), a larger multi-model ensemble will allow a better quantification of the uncertainty and to determine the impact on the modes of climate variability.

*Code availability.* For data retrieval, loading, processing and calculating metrics and visualisation the following R packages have been

used: startR (Automatically Retrieve Multidimensional Distributed Data Sets) version 2.3.0. https://doi.org/10.32614/CRAN.package.startR (Manubens et al., 2023) and s2dv (A Set of Common Tools for Seasonal to Decadal Verification) version 2.0.0. https://doi.org/10.32614/CRAN.package.s2dv (Ho et al., 2023). The scripts used to do the analysis and plots in this manuscript are available at https://doi.org/10.5281/zenodo.14990179 (Bilbao, 2025a).

*Data availability.* The EC-Earth3 decadal predictions are available online: dcppA-hindcast (EC-Earth Consortium (EC-Earth), 2019), dcppC-

hindcast-noAgung (EC-Earth Consortium (EC-Earth), 2022a), dcppC-hindcast-noElChichon (EC-Earth Consortium (EC-Earth), 2022b), dcppC-hindcast-noPinatubo (EC-Earth Consortium (EC-Earth), 2022c), dcpp-EVA (Bilbao, 2025b), dcpp-EVA_H (Bilbao, 2025c) and dcpp-EVA_H_Carn16 (Bilbao, 2025d). ERA5 data is available at https://doi.org/10.24381/cds.adbb2d47 (Hersbach et al., 2020). The Deep-C version 5 reconstructions of the radiation fluxes at the top of the atmosphere are available at https://doi.org/10.17864/1947.271 (Liu et al., 2020). The TLS (Temperature Lower Stratosphere) single-channel data set of zonally averaged temperature anomalies, produced by Remote Sensing

Systems,are available at https://images.remss.com/msu/msu_time_series.html (last access: February 2025) (Mears and Wentz, 2009).



*Author contributions.* Formal analysis: RB. Investigation: all authors. Methodology: RB. Visualization: RB. Writing – original draft preparation: RB. Writing – review and editing: all authors.

*Competing interests.* The authors declare that they have no conflict of interest.

*Acknowledgements.* RB acknowledges support from the European Union's Horizon 2020 research project CONFESS (No 101004156), the HORIZON EU project IMPETUS4CHANGE (No 101081555) and the HORIZON EU project ASPECT (No 101081460). RB and PO acknowledge the support of the Department of Research and Universities of the Generalitat de Catalunya to the CVC Research Group (2021 SGR 00786). TJA acknowledges funding from the Coupled Model Intercomparison Project International Project Office (PO 4000136906) and the European Space Agency (contract number 4000145911/24/I-LR).




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
