# Peer review of "The sensitivity of EC-Earth3 decadal predictions to the choice of volcanic forcing dataset: Insights for the next major eruption."

_EGUsphere, 2025_

## Author Comment (AC1)

**Referee 2:**

Bilbao et al have written a description of experiments run with different volcanic forcing data sets. It is important to be able to quickly generate new volcanic forcings for decadal predictions should a major eruption occur. Therefore, it is great that this work has been done and it is written up in a clear and exhaustive manner. I only have one major comment.

EVA_H has some problems with Agung. For people who might be using EVA_H in the future, can any lessons be learned from how the Agung forcing was produced? See minor comments below. It feels a little like this issue is ignored. If not for Agung and a weak ElChichon peak, EVA_H might be a clear choice over EVA. Can its output be improved? It is clearly capable of hemispherically asymmetrical forcings.

Reply: We thank the reviewer for the positive feedback and valuable comments.

We agree that the possibility to specify a hemispherical asymmetry in EVA but not EVA_H is an important limitation. This issue actually recently came up for the production of the CMIP7 stratospheric aerosol dataset. As a quick fix, we simply spreaded the injection of Agung from the tropic to SH mid-latitudes, but implementing an EVA_style asymmetry factor is an important development required. Other EVA_H improvements have been made for CMIP7 and will be documented in the associated papers, but these were brought forward long after we conducted this work. For the peak forcing of El Chichón, we note that both the $SO_2$ emission that were used in the model and the observations used in GloSSAC are subject to high uncertainty, so we would recommend caution in assessing which model is better there.

**Technical comments:**

L28 aerosols are transported sediment to the troposphere - I don't understand this

Reply: Rephrased.

L108-110 Why use a 95% range for maps, but a 90% range for time series? Not that it matters, as long as you say what you are doing, but it might be better to be consistent.

Reply: The statistical significance of the differences (DCPP-A minus DCPP-C, i.e. volcano minus no volcano) for both time series and fields is computed in the same way. We create a distribution of 10-member mean differences by bootstrapping with replacement, and produce 1000 draws to determine if the 2.5%-97.5% range includes zero. That is consistent with a 95% confidence interval.

Additionally, for the timeseries, the intra-ensemble spread is indicated by showing the 10th and the 90th percentiles. This has been clarified in the text.

L112 If the hindcast drift is state dependent then the drift will be different for those forecasts that contain volcanoes compared to those that do not. Is this worth mentioning?

Reply: Like in previous papers (e.g. Hermanson et al., 2020; Bilbao et al., 2023; Sospedra-Alfonso et al., 2024), we assume that the forecast drift that develops in the

DCPP-A and DCPP-C predictions is the same. Both are initialised from the same initial conditions and have the same radiative forcings, except for the volcanic forcing.

While the precise impact of the volcanic forcing on the forecast drift cannot be determined using three start dates only, comparing the annual global mean temperature climatology (1970-2010) in DCPP-A with the mean of mean of the three start dates s1963, s1981 and s1990 for each prediction experiment (Support Figure 1) shows a similar evolution of the forecast drift, characterised by cooling in the first 4 years. Despite the differences among the experiments, which are due to volcanic forcing and a small contribution from climate noise, the evolution of the drift is similar, which indicates that the impact on the drift is probably small.

[Figure]

[Figure]

*Support Figure 1. a) DCPP-A climatology for the period of 1970-2010 using all start dates. b) Climatology computed using only the s1963, s1981 and s1990 for the hindcast experiments.*

This is consistent with Wu et al. (2023), who produced two hindcasts with and without volcanic forcings and show that the SST drift evolution in the central-eastern tropical Pacific is the same in both with a small offset difference associated with the volcanic forcing (see the Supplementary Figure S9A&B), as the one we can see in Support Figure 1b when comparing EVA/EVA_H with DCPP-A.

L120 Is that not the 90% range? (95-5=90)

Reply: Yes, it has been corrected.

L136 It would also be interesting to discuss why EVA peaks earlier and has a higher peak than EVA_H consistently over the three eruptions.

Reply: The two models were calibrated against different datasets (as detailed in section 2.1), both for emission and aerosol optical properties, and with EVA calibrated against Pinatubo only. This resulted in a longer aerosol production timescale explaining the later peak in EVA_H. It also resulted in greater stratospheric aerosol optical depth (SAOD) for the same mass of $SO_2$ in EVA compared to EVA_H, mostly because EVA_H accounts for both Pinatubo and Cerro Hudson emissions.

L157 Repetition of line 143.

Reply: Removed.

L164 Specify that you are talking specifically about Agung here

Reply: Corrected.

L175 Why did EVA and EVA_H produce such different latitudinal structures for Agung? Could something be done to EVA_H input to improve the outcome?

Reply: Following from the reply at the beginning of the document, while EVA can account for the latitudinal asymmetry with a parameter (indicating the degree of asymmetry in the transport of aerosol to the Northern Hemisphere and Southern Hemisphere), this was not implemented in EVA_H. The hemispheric asymmetry parameter can be estimated for past volcanic eruptions based on observations, however in an operational context the hemispheric asymmetry of a volcanic eruption cannot be anticipated and remains a source of uncertainty.

L181 Again what happened with EVA_H? Why does it not do a descending peak altitude for Agung as it does for the other two eruptions?

Reply: First, we note that although the height of the peak SAOD does not change much for Agung, there is still a shift towards lower altitudes of the extinction values. However, this descend is indeed less clear than for El Chichón and Pinatubo, and the reason is simply the lower $SO_2$ injection height (18 and 20 km for the two phases) whereas for Pinatubo and El Chichón, we used 25 and 28 km respectively. This means that the injection is already relatively close to the tropopause for Agung. We clarified this in the manuscript.

L313 This is wrong, DCPP-C is a "better prediction", please rephrase as it is not clear what is meant here.

Reply: This sentence refers to the Pinatubo eruption in the different experiments (Figure 10l–o). It highlights that in the DCPP-C experiment (Figure 10m), the observed SST in the tropical Pacific lies within the ensemble, unlike in the other experiments. However, we note that this agreement is probably not for the right reasons, as none of the experiments simulate the observed El Niño. Rather, the DCPP-C experiment is warmer due to the absence of volcanic forcing. This evidences the limitations of predicting tropical variability on these 2-5 year timescales.

Line 334 "absence of […] preferential transport to one hemisphere" - this needs to be expanded on. How does this fit with the asymmetrical forcing shown in Figure 2h? EVA_H can clearly do something. This should perhaps be described here or in the introduction.

Reply: The asymmetry (slightly greater AOD values in the NH with respect to the SH) of the EVA_H forcing for the eruption of El Chichón (figure 2h) is consistent with the asymmetry of the eruption of Pinatubo (figure 2i). Similarly, this latitudinal structure of AOD is also present for the eruption of Agung, when in reality it should be more strongly weighted to the SH.

L343 "weaker as it was calibrated accounting for the Cerro Hudson" I thought it didn't have Cerro Hudson and then this was added in an extra experiment? Please clarify.

Reply: EVA_H was not run with Cerro Hudson in our primary experiment, however this eruption (like any other eruption in the MSVOLSO2L4 $SO_2$ emission dataset) was accounted

for when calibrating. This means that without accounting for the Cerro Hudson eruption, EVA_H is not expected to reproduce the post-Pinatubo SAOD as well, in particular with a weaker forcing produced. In addition to this important caveat, there is a difference in the observationally based dataset used in the calibration of these models. As mentioned in section 2.1, EVA was calibrated with the Chemistry-Climate Model Initiative (CCMI, Eyring and Lamarque, 2012) satellite dataset for the 1991 Pinatubo eruption, while for EVA_H used the more recent and improved Global Space-based Stratospheric Aerosol Climatology (GloSSAC). Further details on the calibration can be found in Toohey et al. (2016) and Aubry et al. (2020).

Captions: Many of them have "volc - no volc", while this is probably widely understandable, it is perhaps worth writing it out fully for clarity in case the reader is not familiar with the language used in the community.

Reply: Changed to 'hindcasts with volcanic forcing minus hindcasts without volcanic forcing'.

---

## Author Comment (AC2)

**Referee 1 comments:**

I don't see what is new here compared to another paper just published by this group, Bilbao et al. (2024). Is it just by adding one forcing data set, EVA_H? And since Bilbao et al. (2024) used multiple climate models, why does this paper use only one, and how was it chosen? How model-dependent are the results?

Reply: Bilbao et al. (2024) present a multi-model analysis of the DCPP-C (Boer et al., 2016) decadal predictions, aiming to assess the radiative and dynamical impacts of volcanic eruptions on decadal forecasts, by comparing predictions with/without the official CMIP6 radiative forcings. The objective of this manuscript is different in nature, as it focuses on the uncertainties related to the use of alternative estimations of the volcanic forcings (i.e. from the EVA and EVA_H tools), which as opposed to the CMIP6 forcings, can be derived relatively quickly from a set of observable parameters that describe their main characteristics. In that sense, this study is more closely linked to the multi-model comparison in Sospedra-Alfonso et al. (2024), within the international initiative VolRes, that uses one of these tools to assess the impacts of a hypothetical eruption in 2022, to demonstrate the capacity of operational decadal prediction centers to rapidly respond to future volcanic eruptions. Our paper fills a gap not previously addressed by VolRes, that is the need to validate the forcings generated by EVA and EVA_H tools, in a real (and verifiable) climate prediction context. For this we follow a protocol similar to DCPP-C, by repeating the predictions immediately followed by volcanic eruptions, but using three different forcing estimates: CMIP6, EVA and EVA_H. The decadal predictions are run with EC-Earth3, the climate model developed and used at Barcelona Supercomputing Center.

Regarding the model dependence of the results, Bilbao et al. (2024) showed that the radiative response in the decadal prediction systems analyzed in that study (which used CMIP6 forcing) is largely consistent across models. However, the dynamical responses vary and require large ensembles to be analyzed robustly. Based on these findings, we might expect that predictions using EVA and EVA_H forcings in other forecast systems would yield similar radiative responses, which is the focus of this manuscript. In the summary and conclusions section we already acknowledge the limitation of analyzing dynamical responses with a small ensemble and encourage other prediction centers to conduct similar experiments to support further progress.

Abstracts should not include references.

Reply: The references have been removed.

The abstract has acronyms that are not defined or explained. What is EVA? What is EVA_H? How do they differ? What is BSC? What is CMIP6?

Reply: Acronyms have been spelled out.

There are English errors in the abstract. Did the native English speaker authors not edit the paper?

"Despite these differences, comparing the predicted anomalies in those variables with observations, we show that either of the forcings considered allows to make skillful predictions after the major volcanic eruptions." should be "Despite these differences, comparing the predicted anomalies in those variables with observations, we show that either the EVA or EVA_H forcing would allow skillful predictions to be made after major volcanic eruptions."

Reply: The abstract and the manuscript have been proofread.

I think the last two sentences (after English correction) of the abstract are fundamentally wrong: "Despite these differences, comparing the predicted anomalies in those variables with observations, we show that either the EVA or EVA_H forcing would allow skillful predictions to be made after major volcanic eruptions. Our study thus supports both EVA and EVA_H generated forcings as reasonable choices for predicting the post-volcanic radiative responses."

How do you define "skillful?" How do you define "reasonable?" The problems, as discussed below, are that these forcing data sets cannot be created just after an eruption. The location of the eruption and the satellite-observed $SO_2$ emissions are not enough. Also, the EVA algorithm is too diffusive, and the paper shows that they do not produce the correct latitudinal distribution for the three eruptions studied. And there is no consideration of ENSO and interactions with volcanic eruptions.

Reply: We want to clarify that the location of the eruption and satellite-observed $SO_2$ emissions are known relatively quickly, thus allowing us to create the EVA/EVA_H forcings very soon after the eruptions. For example, EVA_H was run as soon as initial estimates of $SO_2$ mass were communicated through the Volcano Response (VolRes) mailing list after the Raikoke 2019 eruption, which was 1-2 weeks after the eruption (Vernier et al., 2024). Since then, it's been routinely used to estimate the forcing of other eruptions, usually within 2-15 days of eruption occurrence. The forcing estimates can of course be refined once $SO_2$ injection source parameters estimates are refined by the remote sensing community, as was done for Raikoke.

We agree that the latitudinal distributions generated by EVA or EVA_H are simplistic, and that for tropical eruptions transporting aerosol predominantly towards one hemisphere (like Agung 1963 or El Chichón 1982), the predicted latitudinal structure will be too symmetrical. The limitations of these modelling tools are explicitly acknowledged and discussed. However, EVA or EVA_H are not very different from interactive stratospheric aerosol models from that perspective. For example, several models need to spread the Pinatubo injections all the way to the Equator to produce significant dispersion to the northern hemisphere as observed (e.g. Dhomse et al., 2020).

It is also correct that EVA and EVA_H do not account for the impact of the background climate state (including ENSO) on the aerosol forcing process. The limitations of these models in terms of process understanding and their empirical nature is simply a trade-off of their negligible computing cost and simplicity of implementation. Users should carefully consider whether using reduced-complexity models (like EVA or EVA_H) or interactive stratospheric aerosol models is best for their needs. However, given the very large uncertainties in the latter class of models (e.g. Dhomse et al., 2020; Clyne et al., 2021;

Qualia et al., 2023) and the common need for computationally inexpensive approaches (e.g. for rapid response after eruption crisis), reduced-complexity models are valuable tools for numerous applications.

Another important point is that, as far as we know, no other approach to generate more realistic volcanic forcings is suitable for operational prediction purposes, given the short time-constraints to produce them and the fact that both EVA and EVA_H are flexible enough to produce forcings that meet the requirements of any climate model, as already demonstrated in Sospedra-Alfonso et al. (2024). That justifies even further the need of our study, which allows us to understand what aspects of the volcanic response are still realistically represented in the predictions, despite the problems in reproducing hemispherically asymmetric differences in the aerosol loadings.

And why would you want to use either of these forcings after a large volcanic eruption, when models now exist to quickly simulate the conversion of the observed $SO_2$ injections into sulfate aerosols, and the transport and radiative forcing from those aerosols? And these can be updated every month incorporating new observations of how the climate and volcanic aerosol cloud are evolving.

Reply: Current decadal prediction systems contributing to the World Meteorological Organization's Lead Centre for Annual-to-Decadal Climate Prediction do not include interactive stratospheric aerosols, and given that such a major upgrade is unlikely to occur any time soon, these systems will continue to rely on prescribed aerosol fields, at least in the near-term. Stratospheric aerosol models could be run offline to produce the forcing which could be used as input for the models. However, results from previous studies show that interactive aerosol models are unlikely to produce results more realistic than EVA or EVA_H (e.g. Quaglia et al., 2023). These reduced-complexity volcanic forcing models are widely used by the community for numerous applications, including the DCPP and VolRes protocol. As explained previously, their strength resides in their simplicity of use and computationally inexpensive approach, and they are limited by their empirical nature and simplicity. We do not argue that these models are better and just like virtually any numerical modelling community, models covering a wide range of complexity co-exist and complement each other. Our paper merely assesses the performance of tools widely used by the community. We also acknowledge the availability of other models and tools that may produce more realistic forcing distributions, but whose suitability for assisting a multi-model operational exercise is still to be demonstrated.

The use of EVA is very problematic, because how is it possible to know the latitudinal extent of the stratospheric cloud a priori? The 1982 El Chichón and 1991 Pinatubo eruptions were only 2.2° different in latitude, but their clouds ended up centered 15° apart, due to the winds on the day of the eruption. How can the hemispheric asymmetry be known ahead of time?

Reply: The hemispheric asymmetry cannot be known ahead of time, and in fact EVA_H, which does not include this asymmetry, is biased for Agung and El Chichón in terms of latitudinal distribution. In the immediate aftermath of a large tropical eruption, using EVA_H or EVA (in default configuration) would lead to a hemispherically balanced aerosol distribution, as observed for Pinatubo. Indeed, reality might turn out differently, as has been observed for El Chichón and Agung where the spread was hemispherically biased. The

hemispherical asymmetry factor built into EVA does provide some potentially useful possibilities. For example, predictions could be produced for a range of different asymmetries immediately after the eruption, to produce an uncertainty range due to the unknown hemispheric spread. As observations of the aerosol cloud are collected in the weeks-to-months after the eruption, information about the spread of aerosol may be used to better constrain the asymmetry and thus the associated climate prediction. Used in this way, the EVA tools allow for a quantification of uncertainty in the prediction process. In contrast, an interactive aerosol model could not be used in this way, and show strong inter-model spread with, in some cases, quite significant biases compared to observations in terms of the spread of aerosol between hemispheres (e.g., Quaglia et al, 2023, Fig. 2).

Once again, we do not argue that reduced-complexity models are better. Users should always choose modelling tools according to their needs and be aware of the strengths and weaknesses of the tools they use. Our paper simply helps assess those for the EVA and EVA_H reduced-complexity volcanic aerosol forcing models.

In fact, Fig. 2 shows that because of the low latitudinal resolution of the EVAs, the observed hemispheric asymmetry of both 1963 Agung and 1982 El Chichón is not reproduced correctly.  This is an additional problem.

Reply: This is already highlighted in the paper and is not related to a resolution problem. Full-blown models can be equally wrong for tropical eruption, especially if not nudged, and specifying a hemispheric asymmetry in EVA can simply be seen as the reduced-complexity equivalent of nudging in a more complex model.

Some minor issues:

$SO_2$ needs to be spelled with the 2 as a subscript.

Reply: Corrected.

Why is the mass of S used in Table S1 and $SO_2$ in Table S2?  This is confusing.

Reply: The input of EVA is the mass of S (in Tg), while for EVA_H it is the mass of $SO_2$ (in kt). For the paper we have homogenised the units and are now in kt of $SO_2$.

The table headings in Tables S1 and S2 should not have words broken into two lines.  This is easy to fix.

Reply: Corrected.

El Chichón is spelled with an accent mark on the o, but this is not consistent in the text or in the figures.

Reply: Corrected.

Why does Fig. 4 not include observations?  Why are the observations relegated to a different section?

Reply: As explained in the methods section, we refer to climate response as the difference between the predictions with volcanic forcings (DCPP-A, DCPP-EVA and DCPP-EVA_H) and without volcanic forcing (DCPP-C). The difference between the predictions is only the volcanic forcing. The climate response cannot be derived from observations as they include both forced and internal variability signals that cannot be separated from each other. That's why they are not included in the figure.

There were El Niños after each of the eruptions studied. This would have a huge impact on the short term seasonal and annual forecasts, especially after El Chichón in 1982, when most of the cooling from the volcanic eruption was offset by the warming from the El Niño. How was this taken into account?

Reply: As well as driven with external radiative forcings (including the volcanic forcing), decadal predictions are initialised from the observed state, which puts the model in phase with observed internal variability. Decadal predictions have been found to be skillful in predicting ENSO out to several months (e.g. Choi and Son, 2022; Bilbao et al., 2021 (for EC-Earth3)).

What do the "syear" labels on each panel in Figs. 4 and 8 mean?

Reply: "sXXX" refers to the startdate, that is, the date in which the decadal prediction is initialised. For example, "s1990" means the prediction has been initialised in 1990, more specifically in November of 1990 for our model. This is a standard nomenclature in the field of decadal prediction. We now explicitly explain it in the figure caption.

"Notably" and "most notably" and "Note that" are used throughout the paper. What do these mean? They should be deleted. Every sentence should be noted or it should not be in the paper.

Reply: Corrected.

Fig. 6: The parts of the maps that are NOT significant should be hatched, rather than the significant parts. Don't cover the results you want the reader to see.

Reply: Both conventions are widely applied in the climate literature. We nonetheless followed the reviewers suggestion and modified figures 6 and 7 as suggested.

Fig. 6: The temperature changes in the panel labels need units. And why do panels f and i only have one digit after the decimal point, while all the other have two?

Reply: Units are in the caption.

There should be a figure added with the Northern Hemisphere winter responses after each eruption for the first winter, too, to see if the model can reproduce the observed winter warming over Eurasia.

Reply: As stated in the introduction and conclusion, this manuscript focuses only on the global radiative response, that has been shown to be less model-dependent. The winter warming over Eurasia is linked to a dynamic response, which we do not cover as they are more sensitive to the model choice and require a larger ensemble size to be detected. We

acknowledge in the text that volcanic eruptions can lead to dynamic responses, as the example noted by the reviewer. We refer the reviewer to lines 360-371.

The Deep-C version 5 dataset is mentioned several times but never explained. What is it?

Reply: See the methods section (lines 114-115).

References:

Choi, J., Son, SW. Seasonal-to-decadal prediction of El Niño–Southern Oscillation and Pacific Decadal Oscillation. *npj Clim Atmos Sci* 5, 29 (2022). https://doi.org/10.1038/s41612-022-00251-9

Clyne, M., Lamarque, J.-F., Mills, M. J., Khodri, M., Ball, W., Bekki, S., Dhomse, S. S., Lebas, N., Mann, G., Marshall, L., Niemeier, U., Poulain, V., Robock, A., Rozanov, E., Schmidt, A., Stenke, A., Sukhodolov, T., Timmreck, C., Toohey, M., Tummon, F., Zanchettin, D., Zhu, Y., and Toon, O. B.: Model physics and chemistry causing intermodel disagreement within the VolMIP-Tambora Interactive Stratospheric Aerosol ensemble, Atmos. Chem. Phys., 21, 3317–3343, https://doi.org/10.5194/acp-21-3317-2021, 2021.

Dhomse, S. S., Mann, G. W., Antuña Marrero, J. C., Shallcross, S. E., Chipperfield, M. P., Carslaw, K. S., Marshall, L., Abraham, N. L., and Johnson, C. E.: Evaluating the simulated radiative forcings, aerosol properties, and stratospheric warmings from the 1963 Mt Agung, 1982 El Chichón, and 1991 Mt Pinatubo volcanic aerosol clouds, Atmos. Chem. Phys., 20, 13627–13654, https://doi.org/10.5194/acp-20-13627-2020, 2020.

Quaglia, I., Timmreck, C., Niemeier, U., Visioni, D., Pitari, G., Brodowsky, C., Brühl, C., Dhomse, S. S., Franke, H., Laakso, A., Mann, G. W., Rozanov, E., and Sukhodolov, T.: Interactive stratospheric aerosol models' response to different amounts and altitudes of $SO_2$ injection during the 1991 Pinatubo eruption, Atmos. Chem. Phys., 23, 921–948, https://doi.org/10.5194/acp-23-921-2023, 2023.

Sospedra-Alfonso, R., Merryfield, W. J., Toohey, M., Timmreck, C., Vernier, J.-P., Bethke, I., Wang, Y., Bilbao, R., Donat, M. G., Ortega, P., Cole, J., Lee, W.-S., Delworth, T. L., Paynter, D., Zeng, F., Zhang, L., Khodri, M., Mignot, J., Swingedouw, D., Torres, O., Hu, S., Man, W., Zuo, M., Hermanson, L., Smith, D., Kataoka, T., and Tatebe, H.: Decadal prediction centers prepare for a major volcanic eruption, Bulletin of the American Meteorological Society, https://doi.org/10.1175/BAMS-D-23-0111.1, 2024.